# Reliability Improvement of Magnetic Corrosion Monitor for Long-Term Applications

**DOI:** 10.3390/s23042212

**Published:** 2023-02-16

**Authors:** Rukhshinda Wasif, Mohammad Osman Tokhi, John Rudlin, Gholamhossein Shirkoohi, Fang Duan

**Affiliations:** 1School of Engineering, London South Bank University, London SE1 0AA, UK; 2TWI Ltd., Cambridge CB21 6AL, UK

**Keywords:** magnetic eddy current, corrosion monitoring, active redundancy, fault diagnosis, reliability improvement

## Abstract

Electromagnetic techniques are widely employed for corrosion detection, and their performance for inspection of corrosion is well established. However, limited work is carried out on the development and reliability of smart corrosion monitoring devices for tracking internal or buried thickness loss due to corrosion remotely. A novel smart magnetic corrosion transducer is developed for long-term monitoring of thickness loss due to corrosion at critical locations. The reliability of the transducer is enhanced by using a dissimilar active redundancy approach. The improved corrosion monitor has been tested in the ambient environment for seven months to evaluate the stability against environmental factors and degradation. The monitor is found to show great sensitivity to detect defects due to corrosion. Detection of anomalous patterns in the time series data received from the monitors is accomplished by using Pearson’s correlation coefficient. The critical component of the monitor is identified at the end of the test. Research findings reveal that, compared to the existing corrosion monitoring techniques in the industry, the detection and isolation of faulty sensor features introduced in this study can contribute to reliable monitoring of thickness loss due to corrosion in ferromagnetic structures over an extended period of time.

## 1. Introduction

Corrosion is a major concern for asset owners in the energy and power sectors [1]. Huge costs are dedicated to the inspection and maintenance of corrosion in industry [2,3]. According to a recent study, corrosion accounts in some way or another for 42 percent of failures in structures [4]. Thickness loss is among the adverse effects of corrosion in pipes and equipment, deteriorating their performance and integrity. Wall thickness loss due to corrosion can lead to catastrophic consequences if not detected and mitigated in time. Huge economic, environmental, and human life losses are associated with corrosion related incidents in oil and gas industries [5,6,7]. Detection of corrosion at an early stage and continuous monitoring are therefore imperative to avoid these failures.

Monitoring and/or inspecting corrosion is intended to keep track of wall loss over time. Unlike inspections, monitoring involves continuous data acquisition on the state of the component. This is the reason that monitoring is considered better than inspection because it provides continuous information on the status of the component, thereby preventing failures during operation. A number of corrosion monitoring techniques are employed in the industry. These can broadly be divided into two categories (1) Intrusive, and (2) Non intrusive. Intrusive techniques such as weight loss coupons [8,9], electrical resistance (ER) probes [10,11], and electrical impedance spectroscopy [12] involve measurement of corrosion rate by exposing the coupons or probes to the flow conditions [13]. Though intrusive methods are versatile, cost effective and can be applied to all the environments [14], they have high costs and labor requirements associated with the hassle of field rounds, removal and re-installation of probes or coupons in dangerous operating environments, and limited data acquisition [13].

Non-intrusive methods such as ultrasonic [15,16,17], guided waves [18], surface acoustic waves [19], optical fibers [20], and capacitance measurement [21] are installed on the component for corrosion monitoring. Surface acoustic waves [19], optical fibers [20], and capacitance based sensors [21] can only detect surface corrosion and are not capable of detecting buried and internal corrosion defects. Guided wave transducers can screen large pipe lengths [22]. Another major advantage is that they can cover 360° of the pipe circumference as compared to point thickness measurement by UT gauges [23]. However, tests conducted on the stability of the permanently installed guided wave sensors have revealed that the signals are highly affected by the variation in temperature [18]. The most commonly used corrosion monitoring device is the ultrasonic thickness (UT) gauge [24]. There are a number of commercially available UT monitors such as the Rosemount Wireless sensor system, SMS UT system, and WAND system by Inductosense [25,26,27]. The major limitation with UT transducers is the requirement of coupling agents between the component surface and sensors [28]. They need special mechanical devices to be installed in the component surface. They cannot be used on surfaces with paint, coatings, and insulation. Electromagnetic techniques, on the other hand, do not require direct contact and can be used over coatings and paint [29,30]. Due to their simple principle, high sensitivity, and ease of interpretation, these methods are frequently used to inspect corrosion defects. Apart from guided waves, the spatial resolution of the corrosion monitoring devices mentioned above is limited to the area occupied by their sensors. They are designed for tracking wall loss at critical locations, where rates of corrosion are high.

In recent years, corrosion monitoring devices have been developed for steel components. Ha et al. [31] have developed wireless pulsed eddy current sensors for corrosion monitoring of steel framed structures. Zhang et al. [32] proposed a simple MFL sensor using permanent magnet and Hall effect sensor assembly to monitor corrosion in the steel re-bar embedded in concrete. They found that the variation in Hall sensor voltage was directly proportional to the corrosion on the steel. The results were validated using acoustic emission sensors. Zhang et al. [33] developed a micro magnetic sensor to monitor corrosion from the changes in self leakage magnetic flux. Li et al. [34,35] have developed a corrosion monitoring sensor using electromagnet and an array of Hall sensors setup to monitor corrosion in steel re-bars. However, a single NDT technique may not be enough for monitoring corrosion as all NDT methods have some limitations. Therefore, researchers have worked on the development of corrosion monitoring devices with two or more composite sensors. Li et al. [36] developed a novel corrosion monitor using MFL and digital image correlation techniques for monitoring corrosion in steel re-bars in concrete structures. Similar work is carried out in [37], where the MFL and acoustic emission methods are combined to monitor corrosion.

The corrosion monitoring devices mentioned above are for monitoring external corrosion only. The corrosion monitoring of oil and gas pipes requires keeping track of both internal and external corrosion. Limited work is carried out on the development of smart corrosion monitoring devices for internal and buried corrosion. Most of the research is focused on the design of sensors for inspection purpose. Tsukada et al. [38,39] have developed MFL sensors for the detection of internal corrosion in steel structures using magnetic permeability. Long et al. [30] have proposed a magnetic eddy current (MEC) sensor that can work along with MFL sensors in a pipeline inspection gauge to distinguish between internal and external corrosion. The MEC technique is an adaptation of MFL and can be used along with the MFL transducer for corrosion monitoring. While the performance of MFL and MEC methods for detection of corrosion through inspection is well established [30,40], the monitoring devices designed based on the MFL technique are not being tested for long-term performance in the service environment. An improvement in their reliability is required for long-term monitoring applications [41]. It is essential that the transducers withstand variations in the environment and provide reliable data over longer periods of time.

It is challenging to assess the long-term performance and reliability of transducers/sensors for permanent installation [41]. They cannot be calibrated and cross checked frequently as compared to manually operated or temporary sensors [42]. Moreover, regardless of the specialised design and sophisticated maintenance procedures, the sensors will be subjected to failures due to degradation introduced by aging. The failure rate of sensor systems follows a bathtub curve during their life cycle. When detailed knowledge of failure modes for an application is available, reliability can be evaluated by estimating the remaining life by conducting accelerated life tests [43,44,45,46]. Accelerated tests are based on critical failure modes, and by improving design against these critical failure modes, reliability can be enhanced. The wear period of the sensor can be extended by design improvements; however, it cannot be avoided. The reliability of the sensors will eventually decrease with time due to degradation caused by material aging, wear, and corrosion [41]. Degradation of the sensors can lead to either false positives or missed signals. It can also lead to imprecise data due to drift with time. As the aging process is inevitable, advances have been made in the prediction of reliable performance via predictive maintenance [47] and predictive fail-safe [48,49].

The predictive fail safe technique is used more frequently for fault detection in sensors either through analytical [50,51] or hardware approaches [52]. Analytical approaches require prior knowledge about the sensor behaviour, and the service environment to identify and isolate the anomaly of the sensor from the application. This is not always possible, especially for the newly developed sensors. The hardware approach involves using active or passive redundancy to measure the same physical quantity [41]. Multiple sensors, either similar or dissimilar, are installed at the same location for recording measurements. Active redundancy has the benefits of diagnosis of sensor faults such as degradation and/or drift and isolation to ensure reliable performance for long-term application. The faulty sensor diagnosis is carried out by a specialised algorithm [36,53,54] or correlation analysis [55,56] on time series data received from these sensors. Algorithms based on artificial intelligence are more accurate and efficient for complex sensor networks. However, compared to correlation tools, they are complicated and require extensive data from previous experiences for training and learning.

The above discussion reveals that there is very limited work carried out on the development and reliability evaluation of electromagnetic corrosion monitoring devices for both internal and external corrosion. This paper presents the work carried out on the improvement and evaluation of long-term reliability of magnetic corrosion monitor discussed in [57]. The research has the following contributions:Design of a novel smart corrosion monitor based on a dissimilar active redundancy approach to monitor wall loss due to internal and buried corrosion at critical locations;Development of methodology based on Pearson’s correlation coefficient for diagnosis of faulty sensors from time series data;Evaluation of stability and performance of the novel magnetic monitor for an extended time period through real life aging tests conducted for seven months.

The rest of the research paper is structured as follows: Section 2 describes the methodology of the research, and design of the smart corrosion monitor, Section 3 presents the details of the correlation analysis for the identification of faulty sensors, Section 4 illustrates the experimental studies for the evaluation of long-term reliability and validation of the fault diagnosis approach, and Section 5 summarises the conclusions and future work.

## 2. Methodology and Design

The methodology to conduct the research work is presented in Figure 1 and further described in this section.

### 2.1. Design and Development

Both MEC and MFL techniques require a similar excitation circuit to achieve saturation magnetisation in the test specimen. A slight variation in magnetisation level is the only difference between the two methods. However, they work in different ways, so different sensors are needed for detecting corrosion. MEC sensors are harmonically excited coils that measure variation in the magnetic permeability of the test specimen due to wall loss. Magnetic sensors such as Hall effect sensors are used to sense the leakage magnetic field for detection of corrosion in the MFL technique. This characteristic can thus be used for improving reliability by introducing the concept of dissimilar active redundancy. Both sensors can be placed with the same excitation circuit at identical installation points to detect corrosion. The major advantage of active redundancy is that there are very few chances of failure of two sensors at the same time. Furthermore, it is recommended to use different sensors to improve reliability further as different sensors will not have identical flaws [58].

#### 2.1.1. Working Principle

The working principle of MFL and MEC sensors is based on magnetic saturation phenomena. Ferromagnetic materials have a nonlinear relationship between the applied magnetic field and induced magnetic flux owing to their high magnetic permeability. Thus, when an external magnetic field (*H*) is applied by the excitation circuit such as magnetic bridge, the magnetic flux density (*B*) induced in the material is magnified proportionally to the material’s permeability. The relationship between *B* and *H* is expressed as [59]: (1)$$B=\mu_{0}\left(H+H_{m}\right)=\mu_{0}H\left(1+\chi_{m}\right)=\mu_{r}\mu_{0}H$$
where $\chi_{m}$ is the magnetic susceptibility of the material, *H* is the applied magnetic field strength, $H_{m}$ is the magnetisation inside the material, $\mu_{0}$ is the magnetic permeability of free space, and $\mu_{r}$ is the relative magnetic permeability of the material under test. The magnetisation (*B*-*H*) curve and Stoletove (μ-*H*) curve for mild carbon 1002 steel from COMSOL Multi physics material library are shown in Figure 2.

As evident from Figure 2, B increases with increasing H in the region where magnetic permeability is increasing. After a certain point (a), the magnetic permeability starts to decrease owing to the decrease in the capability of the test specimen to carry the magnetic flux. There is an increase in the magnetic flux density (magnetic flux per unit area) called leakage magnetic flux density ($B_{lmf}$) that escapes to the surrounding air. The intensity of the magnetic flux induced in the material and of leakage flux is dependent on the magnetic permeability, which is a function of the material’s magnetic properties. The areas where the wall thickness loss happens are pockets filled with air. Since the magnetic permeability of air is much less as compared to the mild carbon steel, most of the magnetic flux is squeezed in the test specimen, and the magnetic flux density increases from $B_{1}$ to $B_{2}$. Consequently, there is a decrease in the relative magnetic permeability of the test specimen from $\mu_{r1}$ to $\mu_{r2}$ as depicted in Figure 2. However, due to the limited capability of the mild carbon steel to carry magnetic flux close to saturation, there is also an increase in the leakage magnetic flux density from $B_{lmf1}$ to $B_{lmf2}$ above the specimen. Thus, when there is a wall loss due to corrosion, the relative magnetic permeability decreases from $\mu_{r1}$ to $\mu_{r2}$ leading to a proportional increase in the leakage magnetic flux density from $B_{lmf1}$ to $B_{lmf2}$. A schematic representation of the working principle of MEC and MFL is shown in Figure 3.

It can be seen in Figure 3, due to corrosion, that there is variation in both leakage flux above the material specimen and induced magnetic flux in it. This change in induced magnetic flux leads to the change in the magnetic permeability.

The variation in leakage flux can be detected through commercial magnetic field sensors like the Hall effect, giant magneto-resistive (GMR), or tunnel magneto-resistive (TMR) sensors. In MFL applications, Hall effect sensors are considered to be a better option than others due to their high saturation magnetic fields.

Eddy current coils are used in the MEC method to measure changes in the permeability of the test specimen. The impedance of the coil is a function of permeability, conductivity, and coil parameters. The complex part of the impedance is more sensitive to changes in permeability than resistance. The variation in thickness can, therefore, be detected by MFL and MEC sensors fixed in the middle of the magnetisation circuit.

The details on the optimisation of the design of the MEC sensor system and dimensions are discussed in [57,60].

#### 2.1.2. Corrosion Monitor Development

To develop the smart corrosion monitor, Node MCU 0.9 was used due to its proven reliability for long-term applications [61,62]. Node MCU has a built-in ESP8266 System on Chip (SoC) with TCP/IP networking software and microcontroller capability. The 32-bit microcontroller was used to acquire the analog signals for magnetic fields and the 10-bit resolution ADC converted these analog signals to digital for transmission to the data cloud. To measure the reactance, AD-5933 based evaluation board PModia was connected to ESP 8266 micro controller through the I2C interface. The microcontroller was used to program the frequency sweep and store data before transmission. AD5933 is a commercial high-precision impedance analyser chip with a built-in digital signal processing (DSP) engine to carry out frequency sweep [63]. The data acquired from both sensors were sent to the data cloud ThingSpeak through the internet. The block diagram of the smart corrosion monitor is shown in Figure 4.

Node MCU was programmed to acquire the signals from the sensor (MEC) and Hall effect (MFL) sensor with a delay of 50 s to avoid interference between them. To ensure low power consumption, the microcontroller was put into deep sleep mode when not acquiring data. The developed prototype magnetic corrosion monitor is shown in Figure 5.

## 3. Sensor Fault Diagnosis

The data obtained from the sensors have a correlation with the physical world. The multiple sensor systems either use similar or different sensors for the measurement of the same physical parameter at one location. For similar sensors, there exists a correlation between sensor data collected at analogous times and space. In this situation, the sensor data from one sensor have a spatial-temporal correlation. For the determination of the correlation between different sensors monitoring identical parameters, analysis is required to identify anomalies in sensor operation. The individual sensor data may not show any fault, but the correlation analysis can reveal faults when data from different sensors are analysed jointly.

A number of techniques are used to find correlations between multivariate data sets from different sensors such as principal component analysis [64], Kerner principle component analysis [65], and canonical correlation analysis [66]. Pearson’s correlation coefficient or bivariate correlation is used for this research as it is a simple approach to evaluate the strength of the linear relationship between two data sets. Pearson correlation coefficient measures the statistical relationship, or association, between two continuous variables. This method is based on the method of covariance and is considered to be the most effective way to measure associations between variables. It provides information about the magnitude of the association, or correlation, as well as the direction of the relationship. It is independent of the unit of measurement of the variables.

Pearson’s correlation coefficient is calculated using:(2)$$r=\frac{n(\sum xy)-(\sum x)(\sum y)}{\sqrt{\left[n\sum x^{2}-{\left(\sum x\right)}^{2}\right][n\sum y^{2}-{\left(\sum y\right)}^{2}}]}$$
where *r* is Pearson’s coefficient, *n* is the number of readings/measurements, *x* is readings/measurement from the first sensor, and *y* represents readings/measurements from the second sensor

Pearson’s coefficient returns values ranging between −1 and 1. A ‘0’ value indicates no relationship. The higher values suggest a strong correlation. A value in the range [−1, 0] shows a negative relationship between the two variables indicating the movement of the variables in opposite directions. Positive correlation [1, 0] shows that the variables are either increasing or decreasing.

To develop the fault detection scheme, Pearson’s correlation coefficient threshold is defined using training data sets. The training data sets are obtained by information from the sensors from their past history or through tests on the calibration sample with fault-free sensors. In this case, five repeated measurements were taken on mild steel S275JR plates with thicknesses from 3 mm to 10 mm with an increment of 1 mm at each step. The results for the MEC and MFL are presented in Figure 6 and Figure 7 respectively.

As evident from Figure 6, there is a decrease in the reactance of the coil sensor when the thickness of the plate is decreased. For thickness loss of 50% in a 6 mm thick plate, there is a 15% increase in amplitude of the signals. The Hall sensor signals show an inverse trend due to increased leakage magnetic flux. The signals’ amplitude increase by 7% for 50% thickness loss in a 6 mm thick plate.

Pearson’s coefficient values calculated from the five training data sets are listed in Table 1.

Table 1 reveals that a strong negative correlation exists between the readings of MFL and MEC sensors for different plate thicknesses. However, it is important to mention that these data sets were obtained by assuming the corrosion to be uniform. This is not always applicable in the service environment. The working principle of both sensors is different as well as the coverage area. The sensors may have different correlation coefficients in the field where corrosion defects such as sharp pits may affect the signals of both sensors in a different manner. Therefore, a wide range threshold of [−0.5, −1] was considered for distinguishing the sensors functioning without anomaly.

## 4. Experimentation

A real-life aging test was conducted for seven months in an ambient environment to assess the reliability and stability of the improved corrosion monitor. Since corrosion in carbon steel may take years, it is not easy to capture variation in the sensor signals due to corrosion in a few months. Therefore, an experiment was designed to induce corrosion in a pipe section at an accelerated rate. A 6 mm thick mild steel pipe section was used as a test piece. A solar pump was used to circulate salt water to accelerate internal corrosion at the installation locations of prototype corrosion monitors. During the experiment, two prototype sensors were installed on the pipe to detect the wall loss over time. The experimental setup is shown in Figure 8.

The sensors were connected to the internet through a Wi-Fi router to send the signals to the data cloud. The results obtained from both sensors for the period of 1st April–22nd June 2022 from monitor 1 are shown in Figure 9.

It can be seen from Figure 9 that the signals included randomly distributed white noise. A 500-point moving average filter was used to de-noise the signals. The original and filtered signals from the two corrosion monitors are presented in Figure 10 and Figure 11.

At the start of the test, some variation in the signals of both sensors of monitors 1 and 2 was observed. However, due to the deposition of the corrosion products on the pipe, the rate of corrosion was very low. Figure 12 shows the state of the pipe after two and three months, respectively.

The method of salt water circulation for accelerating corrosion was found to be very slow. No significant changes in the signals of sensors were observed. While this demonstrated the stability of sensors in ambient environments, the performance of the sensors to detect corrosion could not be evaluated using this method.

Impressed current is another technique for achieving high corrosion rates quickly. It is widely used to achieve high corrosion rates in steel reinforcement tests for accelerated corrosion in a reasonable time [67,68]. In this technique, corrosion is induced by applying an electric potential between the carbon steel test piece (anode) and the stainless steel bar (cathode). Salt water is used as an electrolyte to enable the flow of current. A schematic representation of the accelerated corrosion by impressed current technique and the test setup for the research is shown in Figure 13.

A voltage potential of 10 V was applied using a DC power supply, and 3% *v*/*v* NaCl solution was used as an electrolyte. The corrosion products were cleaned at regular intervals to achieve high corrosion rates. The test was continued for four months (July 2022–October 2022), and data from both corrosion monitors were recorded on the channels of ThingSpeak data cloud. The results for corrosion monitor 1 are presented in Figure 14.

A considerable variation in both sensors of the monitor was observed. The MEC sensor signals’ amplitude decreased by 20% and MFL sensor signals’ amplitude increased by almost 3%. The MEC sensor was found to be more sensitive to corrosion as compared to the MFL sensor, which can be explained by the difference in the working principle and coverage area of both sensors. The variation in the MFL sensor readings was found to be less as compared to the MEC sensor. Since the area covered by the coil was 20 mm (Outer diameter of the coil) as compared to the MFL sensor (3mm), sharp pits formed away from the MFL sensor coverage area may have resulted in a slight difference in the trend. The signals recorded from corrosion monitor 2 are shown in Figure 15.

The results from the MEC sensor of the second corrosion monitor were different from the expected trend. Pearson’s correlation analysis was conducted on the results obtained from the long-term reliability test to identify the possibility of faults in sensors.

### Identification of Faulty Sensors

The time series data from multiple sensors can be expressed as: (3)$$S\left(t,w\right)=x^{n}+\varepsilon^{n}$$
where *t* is the current timestamp, ω denotes the size of the window, and $x^{n}\left(n=1,2,3,.......\right)$ is the column vector consisting of the nth feature of the signal in the sensor data set for the time interval *t* to t+ω−1. $\varepsilon^{n}$ is the white noise associated with the signal that is randomly distributed. To carry out the correlation analysis for the diagnosis of fault in sensor data, a time slice is defined, and the correlation is calculated using Pearson’s correlation coefficient illustrated in Section 2. In this case, the time window of one month is used. The correlation coefficient for both monitors is presented in Table 2.

Pearson’s coefficient of corrosion monitor 1 shows that both sensors had a strong negative correlation as expected. There are variations observed in the correlation coefficient due to the different working principles and coverage areas. The area covered by the MEC sensor is 20 mm, while the MFL sensor is a point sensor with coverage of a few millimeters. Therefore, a wide range of correlation coefficient thresholds [−0.5, −1] should be considered to overcome this issue. For corrosion monitor 2, the correlation coefficient was found to give an error that indicated the potential problem in the sensor data. This was confirmed by the examination of the corrosion monitors at the end of the reliability test. The impedance evaluation board (PModia) was found to be damaged. The rest of the monitor’s components were in working condition. Node MCU and Hall sensors have been tested extensively and have proven reliability for long-term monitoring applications [61,62,69]. There is not much information on the long-term reliability of the impedance evaluation board (PModia). Moreover, the impedance evaluation board does not have an onboard voltage regulator. This may be one of the reasons for the failure of the board. A study on identifying the failure mode is necessary to be able to improve or redesign the impedance sensor for corrosion monitoring applications in the future.

For validation of the corrosion monitor results, an electromagnetic acoustic transducer (EMAT) thickness probe was used for measurements of the remaining wall thickness on the corroded pipe specimen. A grid of 15 × 15 mm points was used to measure the thickness of the pipe in the areas covered by both corrosion monitors. The test setup is shown in Figure 16.

Thickness measurements obtained from the EMAT probe are presented in Table 3 and Figure 17 and Figure 18.

The thickness map of the pipe areas covered by corrosion monitors 1 and 2 shows variable thickness loss due to corrosion at different spots. At some points, the wall thickness loss was found to be as high as 60% of the pipe’s original thickness (6 mm). The thickness at the location of MFL sensor for corrosion monitor 1 was found to be 4.5 mm. This is the reason the variation in the MFL signals’ amplitude was less compared to MEC sensor signals’ amplitude. Therefore, compared to the training data sets, a difference in Pearson’s correlation coefficient was observed. This can be solved by employing an array of MFL sensors and using a wide threshold for Pearson’s correlation coefficient. The results indicate that sensors were capable of capturing changes in the thickness of the pipe specimen over time.

## 5. Discussion

Improved reliability is vital for sensors developed for long-term applications and permanent installation. The degradation of sensors caused by aging is a natural phenomenon and inevitable. An approach to ensure reliable performance for sensors in condition monitoring is to install multiple sensors for measuring the same physical quantity. As the chances of failure of two different sensors at the same time are very small, improved performance is achieved through fault diagnosis and isolation. A smart novel magnetic corrosion monitor based on dissimilar active redundancy was proposed to improve reliability for long-term applications. A methodology based on Pearson’s correlation coefficient analysis was developed to identify faulty sensors from the time series data obtained from the corrosion monitor.

To evaluate the performance and stability of the corrosion monitor over an extended time, a real-life aging test was designed. A saltwater circulation setup was used to induce corrosion in the mild steel pipe sample. Two prototype corrosion monitors were installed to detect wall loss due to corrosion. The corrosion rate was found to be very slow and there was not much variation in the signals of the sensors received from both corrosion monitors. Compared to the changes in temperature and humidity, the signals of the sensors did not show huge variation. Thus, sensors were found to be stable against environmental factors. However, to capture the changes in the sensor signals due to wall loss, a high corrosion rate was required to complete the test in a reasonable time. The impressed current technique was employed to induce corrosion at an accelerated rate in pipe specimens. Time series data obtained from both monitors were de-noised using a moving average filter, and correlation analysis was conducted. No correlation was found between the sensors of the second corrosion monitor. This indicated a potential fault in one of the sensors. At the end of the test, examination of the components of the corrosion monitor revealed a failure in the AD-5933 evaluation board from the corrosion monitor. Thus, the critical component for the corrosion monitor was found to be the impedance measurement board. A potential cause of failure is the lack of a voltage regulator; however, a detailed study is required to identify the failure mode in the board to improve or redesign for reliability.

The thickness map of the areas covered by the two corrosion monitors was acquired by the EMAT thickness test. The wall thickness loss due to corrosion in the areas was found to be 60% at some locations. This proves the reliability of the developed corrosion monitor for detection of wall thickness loss due to corrosion over an extended time. The study shows that the improved design of corrosion monitor has a better performance, especially where the possibility of cross check and re calibration is not possible. The fault diagnosis and isolation could lead to a better reliability and confidence in long-term corrosion monitoring.

## 6. Conclusions and Future Work

The conclusions from the research carried out to improve the reliability of smart corrosion monitor for ferromagnetic structures are as follows:The signals from the real life aging tests reveal that corrosion monitor based on MFL and MEC sensors have the capability to track wall thickness loss due to internal and/or buried corrosion in an ambient environment. The corrosion monitor was found to not be affected by the variation in temperature and humidity;Pearson’s correlation analysis conducted on the time series data received from the two corrosion monitors was used for identification and isolation of the faulty sensor. This is significant especially for long-term applications with limited access;At the end of the experiments, the critical component of the corrosion monitor is found to be the impedance evaluation PModia board;The study revealed that, compared to the electromagnetic sensors discussed in the literature, the corrosion monitor developed in this study is capable of monitoring wall loss due to internal and buried corrosion remotely. The approach for identification of faulty sensors in the corrosion monitoring device can contribute to a reliable long-term corrosion monitoring in the industry.

A detailed study of the impedance evaluation board PModia is required to identify its failure modes and improve or redesign it in order to improve its reliability. Though PModia is a commercially available board based on an AD-5933 chip, there is no information on the reliability of the board provided by the manufacturers. In the future, the design of the impedance evaluation device especially aimed for corrosion monitoring application should be considered.

The threshold for Pearson’s correlation coefficient was developed based on repeated measurements of plates with uniform thickness losses. In the field, corrosion defects are highly localised. Due to the difference in the coverage area of MFL and MEC sensors, the correlation analysis should consider the effect of different types of corrosion on the sensor’s signal.

This issue can also be solved by using an array of MFL sensors instead of one to develop a better and more accurate correlation for the diagnosis of faulty sensors. The coverage of MFL sensors is limited to the installation point, and the time series data obtained from sensor arrays can give accurate information for correlation analysis. This can also be helpful for corrosion mapping of the area under the corrosion monitor.

The corrosion monitor was designed to give a qualitative indication of wall thickness deterioration over time. Although it is possible to estimate the severity of corrosion from the percentage change in the base signals, it is still necessary to perform UT inspections in order to obtain information about the defect profile. This study can further be extended to investigate methods for quantification of the corrosion defects from the MEC and MFL sensors’ signals. This will help to generate useful data on thickness profiles and corrosion rates.

## Figures and Tables

**Figure 1 sensors-23-02212-f001:**
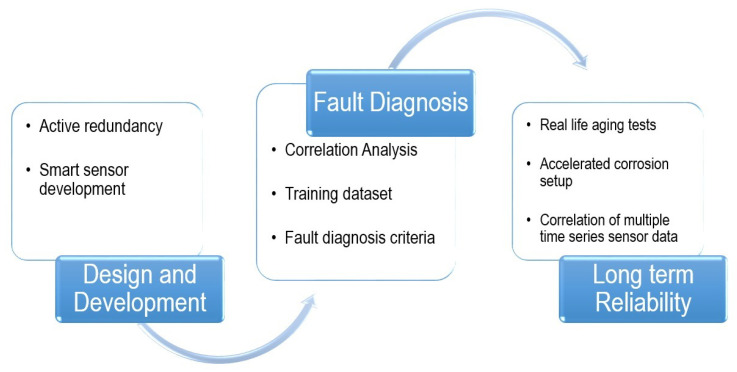
The methodology of the research presented in this paper.

**Figure 2 sensors-23-02212-f002:**
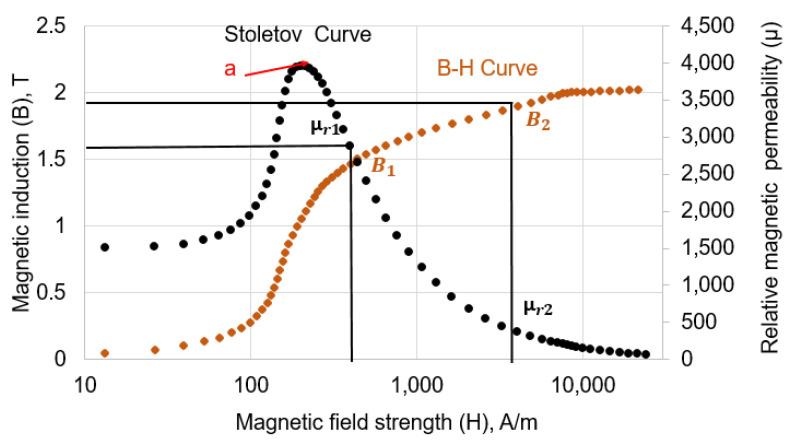
Magnetisation and Stoletov curves of mild carbon steel 1002 showing the relationship between applied magnetic field strength and induced magnetic induction.

**Figure 3 sensors-23-02212-f003:**
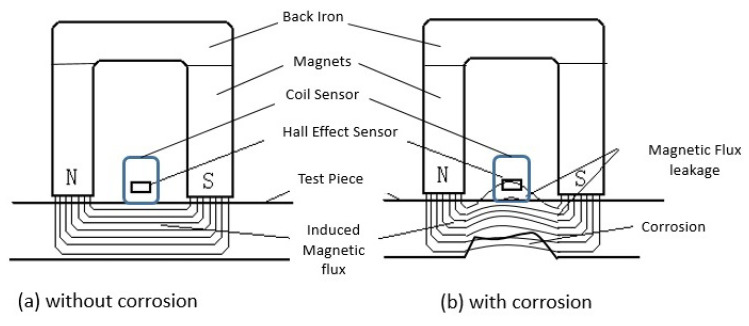
A schematic representation of the MEC and MFL working principles.

**Figure 4 sensors-23-02212-f004:**
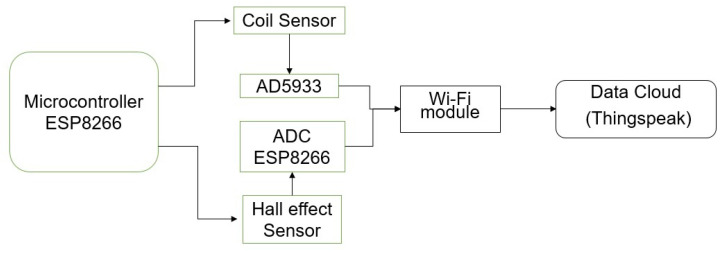
The block diagram of the smart corrosion monitor.

**Figure 5 sensors-23-02212-f005:**
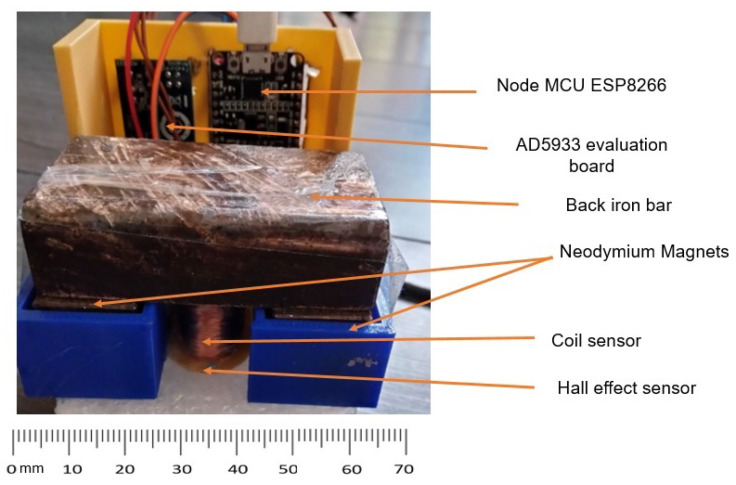
The prototype smart magnetic corrosion monitor.

**Figure 6 sensors-23-02212-f006:**
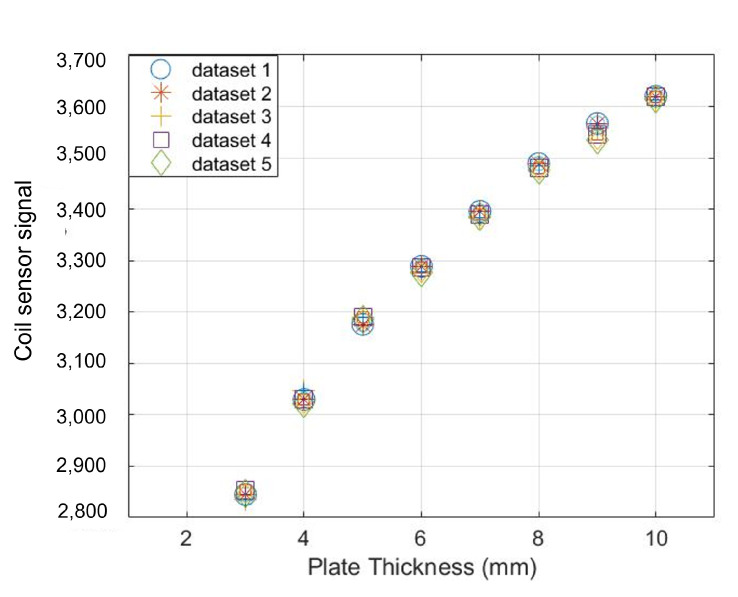
Training data sets for MEC sensors obtained from repeated measurements on mild steel plates for correlation analysis.

**Figure 7 sensors-23-02212-f007:**
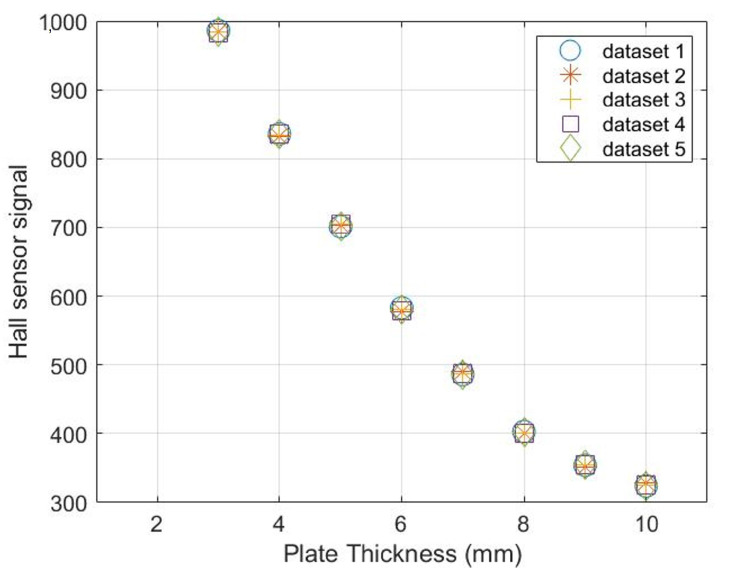
Training data sets for an MFL sensor obtained from repeated measurements on mild steel plates for correlation analysis.

**Figure 8 sensors-23-02212-f008:**
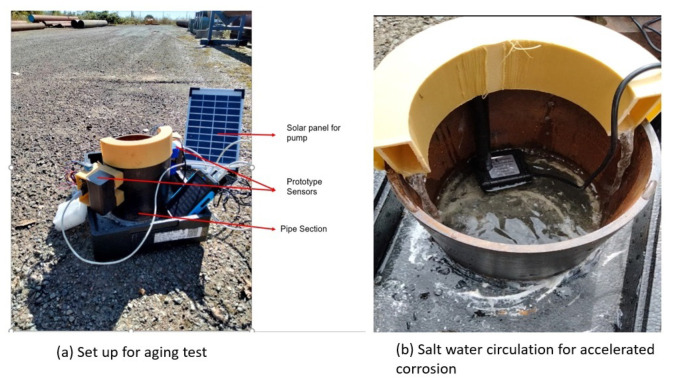
Experimental setup for aging test in an ambient environment.

**Figure 9 sensors-23-02212-f009:**
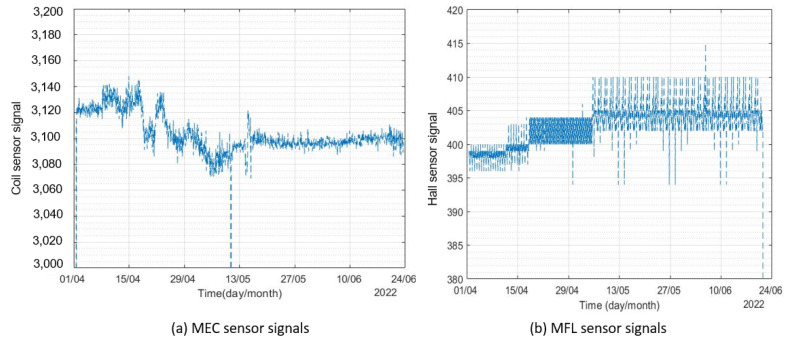
Signals from sensor for the period of April 2022–June 2022.

**Figure 10 sensors-23-02212-f010:**
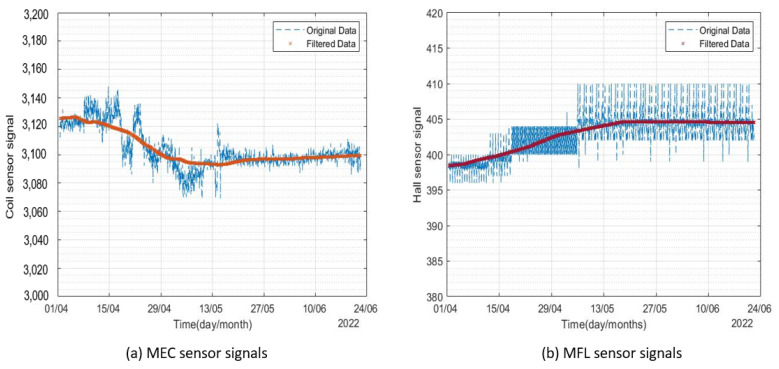
Filtered signals from corrosion monitor 1 for the period of April 2022–June 2022.

**Figure 11 sensors-23-02212-f011:**
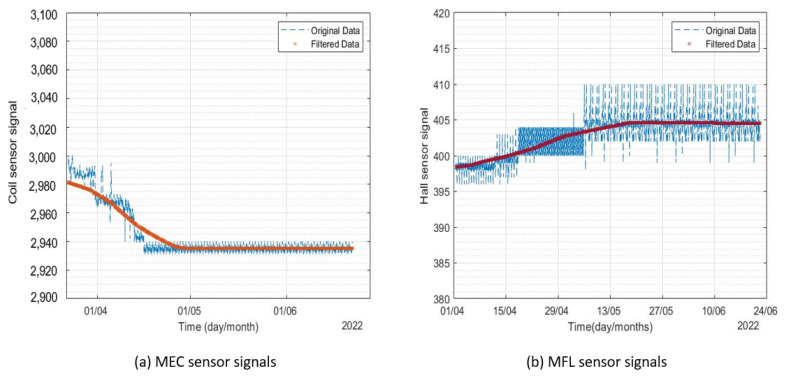
Signals from corrosion monitor 2 for the period of April 2022–June 2022.

**Figure 12 sensors-23-02212-f012:**
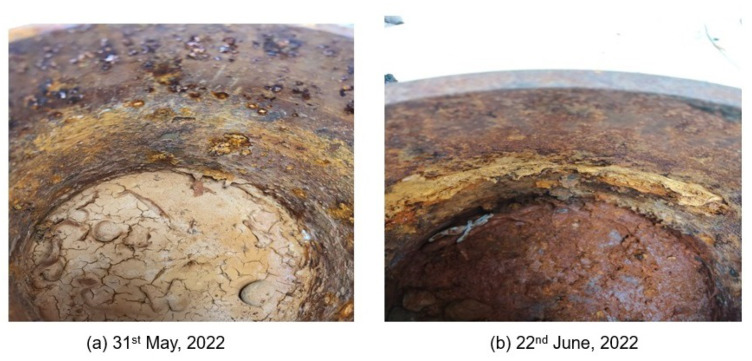
Corrosion on the pipe recorded for the period of April–June 2022.

**Figure 13 sensors-23-02212-f013:**
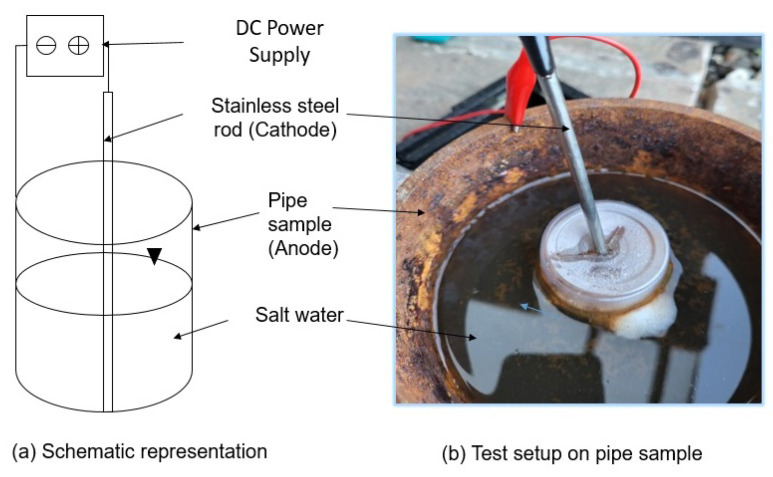
Accelerated corrosion experiment using an impressed current technique.

**Figure 14 sensors-23-02212-f014:**
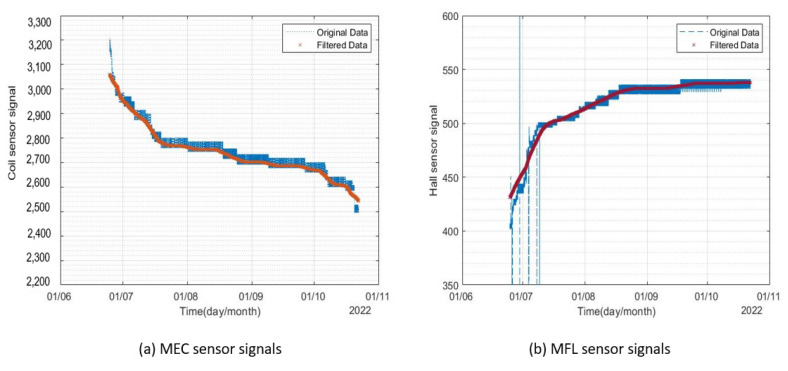
Signals from corrosion monitor 1 for the period of July 2022–October 2022.

**Figure 15 sensors-23-02212-f015:**
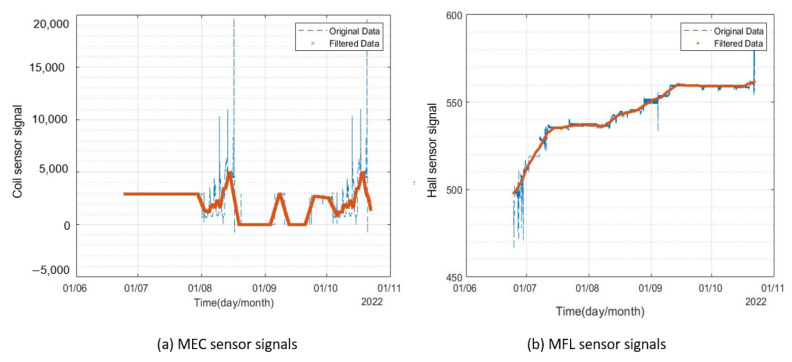
Signals from corrosion monitor 2 for the period of July–October 2022.

**Figure 16 sensors-23-02212-f016:**
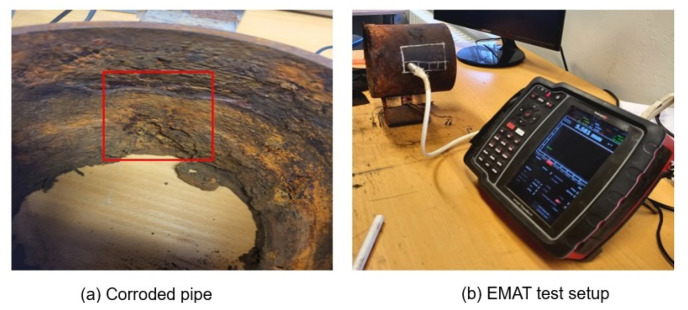
Measurement of thickness of the areas covered by corrosion monitor by the EMAT probe.

**Figure 17 sensors-23-02212-f017:**
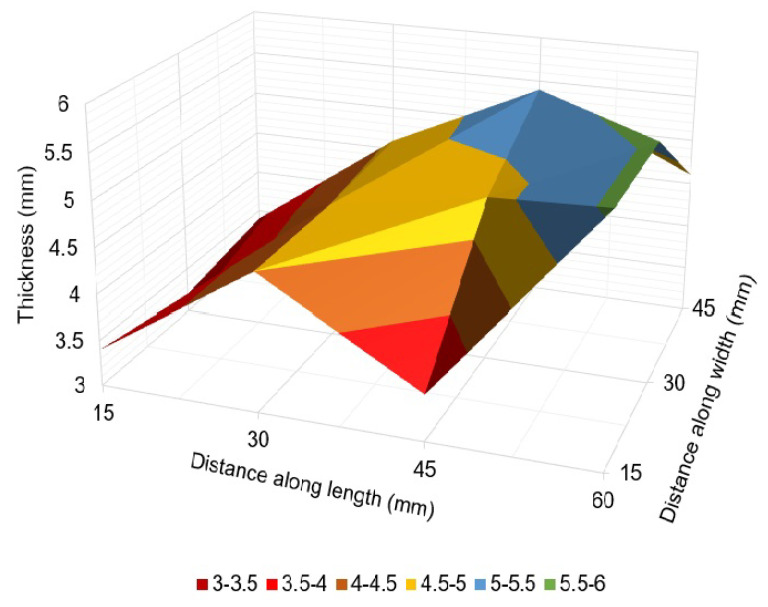
Thickness map of the pipe area covered by corrosion monitor 1.

**Figure 18 sensors-23-02212-f018:**
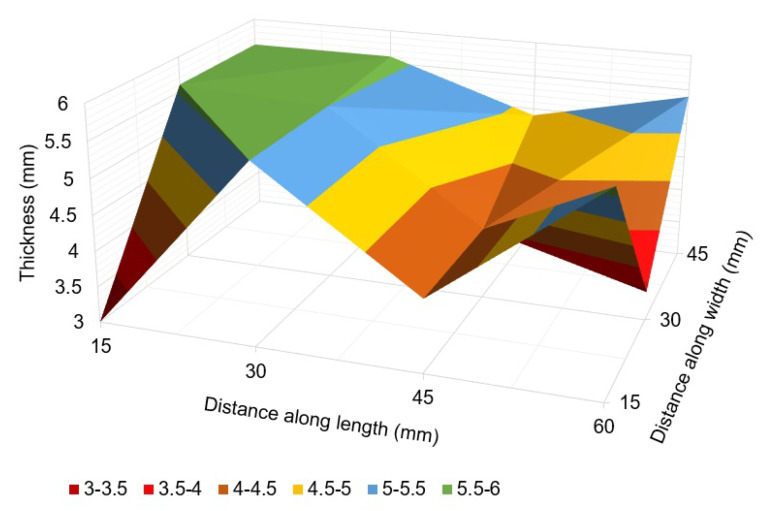
Thickness map of the pipe area covered by corrosion monitor 2.

**Table 1 sensors-23-02212-t001:** Pearson’s correlation coefficient for MFL and MEC sensors calculated from the training data sets.

Dataset No.	Pearson’s Coefficient
1	−0.90
2	−0.92
3	−0.93
4	−0.95
5	−0.89

**Table 2 sensors-23-02212-t002:** Pearson’s correlation coefficient values for the corrosion monitors (* InF = complex square root).

Month	Corrosion Monitor 1	Corrosion Monitor 2
April 2022	−0.95	−0.87
May 2022	−0.87	−0.91
June 2022	−0.94	−0.92
July 2022	−0.94	−1
August 2022	−0.90	InF *
September 2022	−0.85	InF *
October 2022	−0.68	InF *

**Table 3 sensors-23-02212-t003:** Thickness measurements from EMAT for the areas under corrosion monitor 1 and 2.

	Corrosion Monitor 1			Corrosion Monitor 2		
Distance between the magnets (mm)	5	10	15	5	10	15
5	3.4	3.2	3.4	2.5	5.6	5.6
10	4.5	4.5	4.6	5.5	4.3	5.6
15	3.5	4.8	5.4	4	4	4.9
20	5.6	5.6	4.6	5.7	3.4	5.4

## Data Availability

The data presented in this study are available on request from the corresponding author. The data are not publicly available due to privacy and ethical issues.

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
