# Peer review of "Reliability Improvement of Magnetic Corrosion Monitor for Long-Term Applications"

_sensors, 2023, doi:10.3390/s23042212_

Round 1

Reviewer 1 Report

This work studies magnetic sensors for corrosion. The objective is to improve the reliability of long-term corrosion monitoring with such devices.

The manuscript can be accepted for publication only after key points are addressed and explained:

1) How are the measured signals related to corrosion? The corrosion rate? The corrosion extension? The corrosion susceptibility? The corrosion damage (metal remaining after part being lost by corrosion)?

What is the meaning of the measured values? How can we know what is high or low corrosion? For example, from Figures 8, 9, 10, 13 and 14?

2) For a work of this type it is better to use a clean steel (flat sheet is better than a tube) where we can see the corrosion developing and compare with the signals being obtained.

To accelerate the corrosion use acidic solution (or even another metal that corrodes faster, aluminium or magnesium alloys for example). Only when “model” systems in more simple and controlled conditions are tested and validated, we should move to more complex and real systems.

3) Another important aspect, spatial resolution. What is measured is only the region between the magnets? The rest of the material is not under test, correct?

In real application the sensors should mode along the structure? How would it work for a sensor in a fixed location sending signals over time to a remote location?

Just a few more comments:

Section 2.1.1

Page 3, line 114. Magnetic “filed”

(Magnetic) permeability

Relative (magnetic) permeability

Fig 2.

Identify the points in the curves relative to each B1, B2, ur1 and ur2.

Are B1 and ur1 correctly placed? Shouldn’t they have the same H value?

This plot is taken from where? Experimental or some reference?

Page 4, line 124. “Magnetic flux density” is not referred before.

In Figure 3 where is MFL and MEC?

Figure 5. Put a scale bar. What is the brown object in the middle?

Page 7, lines 198, 199: “However, it is important to mention that these datasets were obtained by assuming the corrosion to be uniform.” What does it mean?

Fig 12. Make a better drawing.

Re-check the references. Some are incomplete, others do not seem to be in an acceptable format.

Reviewer 3 Report

In this research, the authors propose a design based on active redundancy for smart magnetic corrosion monitors to improve long-term performance. Here are my comments:

1. It is suggested to be further improved in the abstract, in order to highlight the significance and value of the research.

2. The authors do not provide a detailed description in the INTRODUCTION. Furthermore, for the problems especially solved in this paper, there are few relevant discussions and citations. Please supplement them.

3. The discussion is simple. It is suggested to keep the discussion in a separate section.

4. It is suggested to put the conclusion and future work into a chapter. The conclusion section of the paper lacks thought-provoking observations, elaboration on the directions for further research, and future applications of the presented results.

5. The literature review of fault detection algorithms is not comprehensive enough. The research gap has not been identified clearly. Please comment on and refer to the contribution in the following reference:

[1] Li Z., Jin Z., Gao Y., et al. Coupled application of innovative electromagnetic sensors and digital image correlation technique to monitor corrosion process of reinforced bars in concrete[J]. Cement and Concrete Composites, 2020, 113(7):103730.

[2] Ma D., Fang H., Wang N., et al. Automatic defogging, deblurring, and real-time segmentation system for sewer pipeline defects, Automation in construction. 2022, vol.144, pp.104595, DOI: 10.1016/j.autcon.2022.104595.

[3] Dong J., Wang N., Fang H., et al. Automatic damage segmentation in pavement videos by fusing similar feature extraction siamese network (SFE-SNet) and pavement damage segmentation capsule network (PDS-CapsNet). Automation in Construction, 2022, 143: 104537.

Round 2

Reviewer 1 Report

The minor comments in the reviewer report were addressed and corrected.

The key points in the same report were answered but not included in the manuscript.

I’m afraid this work is of no use for the corrosion community.

The technique is more a way of measuring thickness of the metal wall, not really to measure corrosion.

Author Response

Dear reviewer, 

Many thanks for the comments. Apologies that the answers were not included in the text. We have now improved the text in the paper as follows:

Abstract, line 5, “corrosion at critical locations” is added

Page 1, lines 22-23, “Thickness loss is among the adverse effects of corrosion in pipes and equipment, deteriorating their performance and integrity. Wall thickness loss due to corrosion is added.”

Page 2, lines 59-61, “Apart from guided waves, the spatial resolution of the corrosion monitoring devices is limited to the area occupied by their sensors. They are designed for tracking wall loss at critical locations, where rates of corrosion are high” is added.

Page 3, line 129, “at critical locations” is added.

Page 17, lines 426-430 are included in future works on the determination of corrosion defect profiles for estimating the corrosion size, and corrosion rate.

We agree that the technique discussed is aimed at indicating the wall loss due to corrosion for evaluating the integrity of the assets. However, the technique has the prospect of improvement to estimate the size and rate of corrosion.

Reviewer 3 Report

The authors have eliminated my concerns and answered my questions.

Author Response

Dear reviewer,

Many thanks for taking the time to review our revised paper. We would like to know if there are any further issues with the paper as you have not signed the review report.

Kind regards,

Rukhshinda Wasif
